# Aflatoxin Contamination of Milk Marketed in Pakistan: A Longitudinal Study

**DOI:** 10.3390/toxins11020110

**Published:** 2019-02-13

**Authors:** Agha Waqar Yunus, Nida Imtiaz, Haider Khan, Mohammed Nawaz Mohammed Ibrahim, Yusuf Zafar

**Affiliations:** 1Animal Sciences Institute, PARC National Agricultural Research Center, Park Road, Islamabad 45500, Pakistan; nidaimtiaz234@yahoo.com (N.I.); haiderkhan_1@hotmail.com (H.K.); 2International Livestock Research Institute, ILRI-Pakistan, Park Road, Islamabad 45500, Pakistan; m.ibrahim@cgiar.org; 3Pakistan Agricultural Research Council, G-5/1, Islamabad 44000, Pakistan; y_zafar@yahoo.com

**Keywords:** aflatoxin M_1_, milk, pasteurized, Pakistan, UHT

## Abstract

A longitudinal one-year study was conducted to determine aflatoxin M_1_ levels in different types of milk marketed in Pakistan. Processed and raw liquid milk from 21 sources, two milk powder and six tea whitener brands were sampled on monthly basis from Islamabad. The aflatoxin M_1_ levels in liquid milk were lower (*p* < 0.05) in summer (April to July) compared with the levels in winter (January, November and December). The mean aflatoxin M_1_ levels were 254.9, 939.5, and 1535.0 ng/L in UHT, pasteurized, and raw milk, respectively (differing at *p* < 0.001). The mean toxin level in powdered milk after reconstitution was 522.1 ng/L. Overall, 12.9, 41.0, 91.9 and 50.0% of the UHT, pasteurized, raw and powdered milk samples, respectively, exceeded the Codex maximum tolerable limit of 500 ng of aflatoxin M_1_/L. It was estimated that consumers of raw and processed milk were exposed to 11.9 and 4.5 ng aflatoxin M_1_, respectively, per kg of body weight daily. The study indicates potential aflatoxin M_1_ exposure risks for the consumers of raw milk in the country. The levels of the toxin though comparatively lower in milk powder, requires attention as this type of milk is consumed by infants.

## 1. Introduction

Aflatoxins are the toxic secondary metabolites of various *Aspergillus* spp. that commonly contaminate food and feed ingredients. The aflatoxins encountered in agricultural commodities include aflatoxin B_1_, B_2_, G_1_ and G_2_. In contaminated foodstuffs, the percentage of aflatoxin B_1_ (AFB_1_) in total aflatoxins is over 90%. Once ingested by animals, AFB_1_ is also carried to milk in the form of the toxic metabolite aflatoxin M_1_ (AFM_1_) [1]. All of these toxins are known to exert potent hepatotoxic, immunotoxic and carcinogenic effects in animals and humans consuming the contaminated food. Due to high carcinogenicity, aflatoxins are the only group of mycotoxins for which legislation and control protocols are in place, even in many developing countries [2]. The toxicity of aflatoxins is known to be higher in younger age groups (infants, children and young animals). Monitoring the levels of AFM_1_ in milk and baby foods is therefore more critical. Consequently, the levels of the mycotoxin allowed in milk are lower than the levels allowed in other foodstuffs. The EU further restricts the levels allowed in infant milk formula to half of the levels allowed in milk.

The maximum tolerable limit of AFM_1_ in liquid milk is 500 ng/L in the USA and in the Codex standards, while only 50 ng/L in the EU. In Pakistan, the maximum tolerable limit of AFM_1_ is 10 µg/kg in milk powder while no particular legislation has been made for liquid milk. This is despite the fact that specific monsoon conditions in the country favor mycotoxin development in food and feedstuffs, pushing Pakistan into a high risk area. The studies conducted in Pakistan also show that 25 to 90% of milk samples [3,4,5,6] could be contaminated with AFM_1_.

There have been notable differences in the AFM_1_ levels in milk reported by different authors from Pakistan. In this regard, Muhammad et al. [4] reported 17,380 ng/L as the mean AFM_1_ level in milk sampled from Lahore in the year 2007, with 81% samples exceeding the 500 ng AFM_1_/L limit. Contrary to this, Iqbal et al. [6] reported 64 ng/L mean AFM_1_ level in milk sampled in the year 2011 in the urban areas of Punjab province, with 15% samples exceeding the 500 ng AFM_1_/L limit. These differences in the AFM_1_ contamination level reported by various authors could be due to different seasons, different feeds used by farmers in different areas, and different methods of AFM_1_ quantification. Overall, such differences make it impractical to infer risk of exposure for the consumers of milk in other cities. The present study was therefore conducted as a longitudinal one year study to determine the AFM_1_ levels in various types of milk, primarily processed, available in Islamabad the capital city of Pakistan. To the best of our knowledge, there is no previous longitudinal study on AFM_1_ contamination in processed milk in Pakistan. Also, AFM_1_ contamination of milk has not been previously investigated in Islamabad city. Data on processed milk from one city are however applicable to milk consumers in other cities because processing companies collect milk from farmers located in different areas and distribute it to consumers in all cities of Pakistan.

## 2. Results and Discussion

### 2.1. Aflatoxin Contamination of Liquid Milk

The seasonal trend regarding aflatoxin contamination of liquid milk (average of UHT, pasteurized, and raw milk) is presented in Figure 1. The levels of AFM_1_ in liquid milk were lower (*p* ≤ 0.021) in the summer months of April to June compared with the levels in the winter months of January, November and December. This seasonal variation in AFM_1_ levels is in accordance with previous findings in which comparatively lower levels of AFM_1_ were found in raw milk sampled during summer months in the country [7,8].

The data regarding AFM_1_ levels in different types of liquid milk are presented in Table 1. In case of UHT milk, the mean AFM_1_ level was 254.9 ng/L, with 12.9% samples exceeding 500 ng/L, the maximum allowed limit in the USA. The maximum AFM_1_ level noted was 1536 ng/L. Only five out of 85 UHT milk samples (5.8%) had AFM_1_ levels less than 50 ng of AFM_1_/L, the maximum allowed limit in the EU. Overall, the UHT milk samples had lower (*p* < 0.001) AFM_1_ levels compared with the pasteurized and raw milk samples.

In case of pasteurized milk, the mean AFM_1_ level was 939.5 ng/L and only 41% samples were below the limit of 500 ng AFM_1_/L. Only four pasteurized milk samples (5.1%) were below the limit of 50 ng AFM_1_/L. Compared with the UHT milk, the pasteurized milk had higher (*p* < 0.001) AFM_1_ levels. Raw milk was found to have higher (*p* < 0.001) AFM_1_ levels than the UHT and pasteurized milk. None of the 34 raw milk samples qualified the EU limit of 50 ng AFM_1_/mL milk. Only five raw milk samples (9.1%) had AFM_1_ levels lower than the USA standard of 500 ng/L of milk, indicating potential health risks for the consumers of raw milk. Mean AFM_1_ level in raw milk in the country has been reported to be in the range of 46 to 340 ng/L by various authors, which is lower than the mean level being reported here [5,8,9]. However, the mean AFM_1_ level in raw milk collected from peri-urban farms in Lahore was found in one report to be 17,380 ng/L which is very high compared to the present findings [4]. The differences in various reports indicate that the AFM_1_ contamination may vary in different seasons and areas. 

In general, the milk sampled from rural areas of Pakistan, where more green fodder is available, has been found to have lower AFM_1_ levels compared to the milk collected from urban areas [6]. This could also explain the lower levels found in UHT milk in the present study which is in general collected from rural areas followed by testing for AFM_1_ levels before packaging. The farmers are usually paid according to the quality of the milk. The UHT milk companies also have the option to either not buy or discard the milk having higher AFM_1_ contamination. Compared to the UHT milk, the companies marketing pasteurized milk have their own dairy farms and thus do not have the option to discard milk. It is therefore important to educate these companies to control aflatoxin in the dairy feeds, coupled with legislation, for a market advantage.

### 2.2. Aflatoxin Contamination of Tea Whiteners and Milk Powder

The data regarding AFM_1_ levels in tea whiteners are presented in Table 2. The mean AFM_1_ level in tea whiteners was only 98.9 ng/L, with 2.1% samples (1 out of 48) exceeding the 500 ng AFM_1_/L limit. The mean AFM_1_ levels in different months did not differ statistically (*p* = 0.081). Tea whiteners are usually made using both the milk and non-milk constituents, hence it is logical to expect a lower AFM_1_ contamination in this commodity. It should however be noted that tea whiteners are not a substitute of milk, and the present results do not imply use of such products as a replacement of milk.

The mean AFM_1_ level in powdered milk after reconstitution was 522.1 ng/L i.e., close to the maximum tolerable limit of 500 ng AFM_1_/L. In total, 50.0% of the powdered milk samples had AFM_1_ levels lower than the 500 ng/L limit. Only two powder milk samples (10.0%) passed the EU standard of 50 ng AFM_1_/L. It appears from these results that the powdered milk was a safer option for consumers compared to the pasteurized and raw milk. However, these results need to be examined in consideration of the fact that the studied milk powder brands are used as infant milk formula. As infants are more sensitive to the hazards of AFM_1_, effective measures should be in place to ensure that all powdered milk brands marketed in Pakistan meet at least the 500 ng AFM_1_/L limit.

### 2.3. Miscellaneous Quality Variables of Milk Samples

Miscellaneous quality variables were studied to identify any case of adulteration which would result in dilution of the natural AFM_1_ content of milk and making comparison biased. Data regarding fat, total solids, and SNF content of milk samples are presented in Table 3. Fat percentage was lower (*p* ≤ 0.029) in the UHT milk compared with the pasteurized milk samples in the month of January. The UHT milk samples were also found to have lower (*p* ≤ 0.037) total solids compared with both the pasteurized and raw milk samples in January. These data are indicative of fat removal by milk processors to make milk uniform regarding fat content. The SNF contents of different types of milk did not differ during the study duration. Furthermore, no addition of starch, cane sugar, urea, detergent and hydrogen peroxide was found in liquid milk samples. These data indicate that no particular adulteration occurred in the liquid milk samples included in the study.

In the case of tea whiteners, one brand was found to be positive for neutralizer/soda in April and then from June through December. Another brand of tea whitener was found to be positive for detergent only in January. 

### 2.4. Exposure to Aflatoxin through Milk

The users of raw and processed milk (UHT and pasteurized) were estimated to be exposed to 11.9 and 4.5 ng AFM_1_ per kg body weight daily, respectively (Table 4). The overall daily AFM_1_ intake was found to be 11.6 ng per kg body weight, which is very high compared with 3.1 ng/kg mean estimated previously for consumers in Karachi [10]. The higher exposure estimated in the present study could be due to the higher contamination levels noted in Islamabad, and the use of weighted AFM_1_ mean in our study. The mean AFM_1_ in milk if not corrected for relative consumption would have resulted in an estimate of 6.4 ng/kg day.

The presently estimated AFM_1_ intake for children using milk powder is alarmingly high despite the acceptable levels of AFM_1_ in milk powder for one-year-old children. Compared to adults, the consumption of milk per kg body weight is higher in case of children which results in higher contribution of aflatoxin from milk. It may be noted that UHT milk was found to have the lowest AFM_1_ levels in the present study. If UHT milk was used for calculation of daily exposure of children, then the figures for one- and five-year-old children had been 13.8 and 6.9 ng per kg body weight. This shows the need to reduce the levels of AFM_1_ in milk powder produced in the country, especially in the milk for the five-year age group. In a separate study (unpublished data), we found that the AFM_1_ levels in imported milk powders was less than 60 ng/L and would result in an exposure of only 3.2 ng/kg body weight daily in one-year-old children. This implies that during legislation, the levels of AFM_1_ allowed in milk for one-year-old and younger children should be kept lower than 500 ng/L.

## 3. Conclusions

This study, though conducted on a limited scale, identifies raw milk as less suitable for human consumption in the capital city Islamabad due to high AFM_1_ levels. The processed liquid milk was found to be a safer option for consumers. In all types of milk, the levels of AFM_1_ were lower during summer months. Reducing the levels of AFM_1_ in milk for infants, while strictly following the 500 ng AFM_1_/L limit in milk for older children are suggested as ways to reduce the exposure of children to AFM_1_.

## 4. Materials and Methods

### 4.1. Sampling

All the regularly available locally produced processed milk brands were collected once a month from supermarkets in the twin cities of Islamabad-Rawalpindi in the year 2016. Each time, 21 different liquid milk samples were collected including eight UHT milk brands, seven pasteurized milk brands, and raw milk from six different suppliers. Additionally, two brands of milk powder—each for a different age group of children—and six brands of tea whiteners were sampled on monthly basis.

As UHT and powdered milk have long expiry time, it was ensured that a different and fresh batch was collected on each sampling. The collected liquid milk samples were transported in chilled form to the laboratory where these were subsampled and stored at −20 °C until AFM_1_ analysis. Identity of the collected samples was masked from analysts to make the study blind.

### 4.2. Aflatoxin M_1_ Analysis

The milk samples were analyzed for AFM_1_ contamination using ELISA kits (AM1-E01, Immunolab GmbH, Kassel, Germany) following protocols specified by the manufacturer. The kit had a quantification range of 10 to 1000 ng of AFM_1_/L. Samples were analyzed several times in different dilutions until the AFM_1_ levels in the diluted samples fell within the quantification range of the ELISA kit. Limit of detection (LOD) was found to be 4.4 ng AFM_1_/L, while recovery of AFM_1_ was 86.9% at 500 ng of AFM_1_/L. Results were not corrected for recovery.

The kit was validated before start of this study [13]. In addition, external standards were run with each microtiter plate to ensure reliability of the AFM_1_ quantification. Two types of reference materials were used to make external standards. The first was a 44 ng of AFM_1_/L reference skim milk powder (RMBD-248, EU Joint Res Center, IRMM, Geel, Belgium) which was used to make 4.4 and 44 ng of AFM_1_/L standards. The second reference material was a purified 9.786 µg of AFM_1_/mL solution (46319U, Supelco, Bellefonte, PA, USA) which was diluted with ddH_2_O to make 5, 50, 500 ng of AFM_1_/L secondary standards for use with each microtiter plate.

The ELISA kits were read on an ELISA reader (BDSL Immunoskan MS 355, Labsystems, Vantaa, Finland). For quantification of AFM_1_, a software based on four parametric curve estimation provided by the kit manufacturer was used.

### 4.3. Miscellaneous Milk Quality Variables

Raw milk in the country is often adulterated with water, urea, starch, and neutralizers, etc [14]. Comparison of adulterated milk with pure milk is not justified because addition of water and non-milk constituents result in milk dilution and lowering of the AFM_1_ contents. Therefore, all the liquid milk samples were analyzed for milk fat, total solids, and solids-not-fat (SNF) to identify any notable case of adulteration. Milk fat was determined following the Gerber method and using a Gerber Centrifuge (Funke Dr. N. Gerber, Berlin, Germany) [15]. Total solids were determined using Quevenne lactometer and following the principles of lactodensitometry as detailed by Ling [16]. Solids-not-fat were calculated as difference between total solids and milk fat.

Qualitative tests were performed to detect adulterants and stabilizers including starch, cane-sugar, urea, detergent, and hydrogen peroxide following methods described by Sharma et al. [17].

### 4.4. Calculations and Statistics

The exposure to AFM_1_ through a specific milk type was estimated using a formula:

Average daily AFM_1_ intake (ng/kg body weight) through a specific type of milk = [(mean AFM_1_ ng/L in the specific type of milk) × (milk availability per head per day in Pakistan)]/(average body weight)
(1)
where, milk availability was taken as 170.2 L/head/year [11], and average body weight was taken as 60 kg. Daily consumption of milk powder in case of children was taken as recommended by the milk powder manufacturer. To calculate the exposure level for one and five year old children, average body weight was taken from Royal College of Pediatrics and Child Health (UK-WHO) charts.

In case of overall AFM_1_ intake through liquid milk, the mean AFM_1_ levels were normalized using a formula, keeping in view only 5% share of processed milk in the overall milk market:

Average daily AFM_1_ intake through liquid milk (ng/kg body weight) = [(mean AFM_1_ in raw milk × 95) + (mean AFM_1_ in processed milk × 5)]/100
(2)

Means were compared using ANOVA and LSD in IBM SPSS Statistics 20 (IBM Corp., Armonk, New York, NY, USA, 2011). Values of AFM_1_ less than the LOD were considered as zero for calculating monthly means.

## Figures and Tables

**Figure 1 toxins-11-00110-f001:**
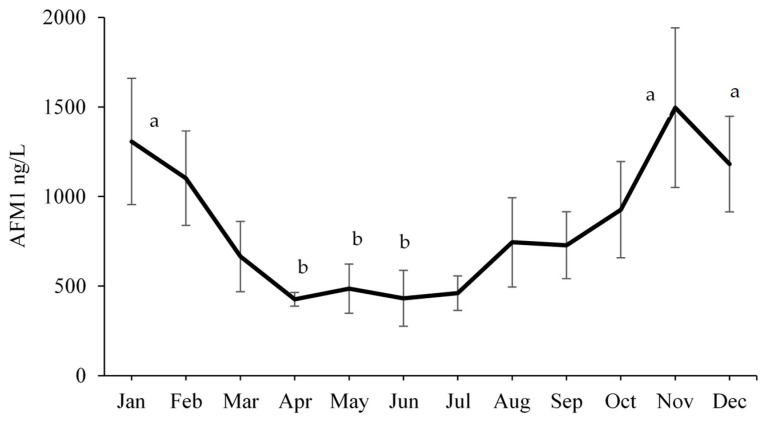
Average aflatoxin M_1_ levels ± SEM, in liquid milk during different months (means bearing different alphabets differ significantly at *p* < 0.05).

**Table 1 toxins-11-00110-t001:** Aflatoxin M_1_ levels (ng/L) ^1^ in liquid milk samples during the year 2016.

Month	UHT Milk	Pasteurized Milk	Raw Milk	*p* Value	*F* Value
Jan	434.0 ± 517.5 ^b^	1861.1 ± 1855.9 ^ab^	2422.3 ± 198.6 ^a^	0.043	4.12
Feb	135.5 ± 34.4 ^b^	1613.9 ± 905.0 ^a^	2155.7 ± 603.9 ^a^	0.000	18.25
Mar	48.3 ± 37.8 ^c^	556.0 ± 476.5 ^b^	1908.7 ± 367.3 ^a^	0.000	40.32
Apr	413.2 ± 201.2	334.0 ± 151.6	549.2 ± 38.3	0.055	3.49
May	109.5 ± 48.2 ^b^	553.8 ± 770.5 ^ab^	994.6 ± 539.7 ^a^	0.029	4.41
Jun	61.4 ± 26.1 ^b^	355.2 ± 351.1 ^b^	1123.3 ± 1015.2 ^a^	0.022	5.07
Jul	213.5 ± 150.4	565.3 ± 666.6	628.3 ± 134.4	0.169	1.98
Aug	218.9 ± 82.1	1148.3 ± 1820.0	886.2 ± 289.0	0.289	1.34
Sep	278.2 ± 124.1	813.8 ± 1073.7	1365.9 ± 459.2	0.078	3.04
Oct	279.5 ± 99.5 ^b^	913.3 ± 1238.3 ^b^	2231.7 ± 847.1 ^a^	0.008	6.89
Nov	441.1 ± 184.8 ^b^	1385.6 ± 1469.5 ^b^	3796.3 ± 2858.1 ^a^	0.009	6.40
Dec	395.1 ± 155.5 ^b^	1570.9 ± 1457.5 ^a^	1894.4 ± 1184.4 ^a^	0.040	3.92
*n*	85	78	55		
Mean	254.9 ± 223.9 ^c^	939.5 ± 1164.6 ^b^	1535.0 ± 1234.5 ^a^	0.000	31.74
Median	198.4	368.5	1037.6	-	-
Max	1536.0	4808.0	7460.7	-	-
Min	n.d.	32.8	1912	-	-

^1^ Arithmetic means ± standard deviation. ^abc^ Means bearing different superscripts differ significantly within a row (*p* < 0.05). n.d. = less than detection limit.

**Table 2 toxins-11-00110-t002:** Aflatoxin M_1_ levels (ng/L) in tea whiteners.

Month	Tea Whiteners
Jan	138.8 ± 195.7
Feb	49.4 ± 83.3
Mar	0.6 ± 0.9
Apr	178.2 ± 173.5
May	16.4 ± 6.2
Jun	32.1 ± 24.4
Jul	69.4 ± 89.3
Aug	120.2 ± 55.4
Sep	60.9 ± 40.7
Oct	38.5 ± 30.5
Nov	120.8 ± 80.8
Dec	339.4 ± 357.3
*p* value	0.081
*F* value	1.849
*n*	48
Overall Mean	98.9 ± 161.8
Median	40.6
Max	932.6
Min	n.d.

Data presented as arithmetic means ± standard deviation; n.d. = less than detection limit.

**Table 3 toxins-11-00110-t003:** Quality variables of liquid milk included in the study.

Mon	Fat (%)				Total Solids (%)				SNF (%)			
	UHT	Pasteur	Raw	*p*	UHT	Pasteur	Raw	*p*	UHT	Pasteur	Raw	*p*
Jan	3.21 ± 0.35 ^b^	4.12 ± 0.13 ^a^	3.84 ± 0.69 ^a^	0.014	11.00 ± 1.22 ^b^	12.56 ± 0.18 ^a^	12.22 ± 1.02 ^a^	0.027	7.78 ± 0.96	8.44 ± 0.23	8.38 ± 1.46	0.498
Feb	3.49 ± 0.29	4.12 ± 0.13	3.86 ± 0.64	0.077	11.62 ± 0.93	12.50 ± 0.28	11.79 ± 1.39	0.361	8.13 ± 0.69	8.37 ± 0.31	7.93 ± 1.48	0.758
Mar	3.66 ± 0.38	4.58 ± 0.13	3.90 ± 0.90	0.061	11.73 ± 0.87	12.31 ± 0.34	12.11 ± 0.62	0.331	8.07 ± 0.78	7.72 ± 0.37	8.22 ± 1.03	0.591
Apr	3.58 ± 0.23	3.72 ± 0.38	3.59 ± 0.69	0.868	11.66 ± 0.63	12.00 ± 0.45	11.83 ± 1.18	0.808	8.08 ± 0.57	8.27 ± 0.26	8.25 ± 1.24	0.911
May	3.36 ± 0.46	3.90 ± 0.28	4.09 ± 0.85	0.103	11.49 ± 0.98	12.00 ± 0.72	12.53 ± 0.89	0.106	8.12 ± 0.61	8.08 ± 0.51	8.44 ± 1.45	0.778
Jun	3.46 ± 0.27	3.76 ± 0.50	4.01 ± 0.94	0.302	11.73 ± 0.68	12.21 ± 0.62	12.26 ± 0.87	0.364	8.24 ± 0.44	8.45 ± 0.28	8.25 ± 1.29	0.914
Jul	3.63 ± 0.63	3.96 ± 0.44	4.14 ± 1.06	0.484	11.78 ± 1.16	12.37 ± 0.35	12.21 ± 1.04	0.547	8.15 ± 0.71	8.40 ± 0.33	8.07 ± 1.37	0.838
Aug	3.36 ± 0.18	3.10 ± 0.97	3.88 ± 0.61	0.100	11.58 ± 0.85	11.25 ± 1.33	11.97 ± 1.15	0.513	8.21 ± 0.67	8.15 ± 0.53	8.10 ± 1.41	0.978
Sep	3.56 ± 0.32	4.00 ± 0.45	4.01 ± 0.91	0.336	11.84 ± 0.93	12.53 ± 0.55	12.19 ± 1.34	0.548	8.38 ± 0.65	8.53 ± 0.23	8.15 ± 1.68	0.848
Oct	3.52 ± 0.22	3.96 ± 0.53	3.97 ± 1.14	0.494	12.59 ± 0.70	13.15 ± 0.78	12.56 ± 1.12	0.484	9.07 ± 0.60	9.19 ± 0.25	8.58 ± 1.28	0.438
Nov	3.54 ± 0.29	4.00 ± 0.71	4.16 ± 0.80	0.193	11.18 ± 1.55 ^b^	12.98 ± 0.85 ^a^	12.10 ± 0.72 ^ab^	0.040	7.64 ± 1.41	9.00 ± 0.24	7.93 ± 0.87	0.090
Dec	3.41 ± 0.34	3.76 ± 0.32	4.04 ± 0.63	0.069	11.46 ± 1.18	11.81 ± 0.56	12.04 ± 0.79	0.474	8.04 ± 0.99	8.05 ± 0.49	8.01 ± 1.19	0.996

Data presented as arithmetic means ± standard deviation; UHT = milk with ultraheat treatment, Pasteur. = pasteurized milk. ^abc^ Means bearing different superscripts differ significantly within a row under each quality variable (*p* < 0.05).

**Table 4 toxins-11-00110-t004:** Estimated daily AFM_1_ intake in users of different types of milk.

Type of Milk	AFM_1_ (ng/L)	Daily Milk Consumption ^1^ (mL)	Average Body Weight ^2^ (kg)	Daily Intake (ng/kg bw)
Children:				
Milk powder for 1 year old	353.1	500	9.3	19.1
Milk powder for 5 year old	728.7	500	18.5	19.7
Adults:				
UHT	254.9	466	60.0	2.0
Pasteurized	939.5	466	60.0	7.3
Raw	1535.0	466	60.0	11.9
Overall liquid	1487.4	466	60.0	11.6

^1^ Milk consumption for children as recommended by the milk powder manufacturer. Milk consumption in case of adult is the per head milk available in Pakistan [11]. ^2^ Average body weight of children taken from Royal College of Pediatrics and Child Health (UK-WHO) charts. Average weight of Pakistani children of one year and five years of age are 5.6 and 15.3, respectively [12]. bw = body weight.

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
