# Peer review of "Aflatoxin Contamination of Milk Marketed in Pakistan: A Longitudinal Study"

_toxins, 2019, doi:10.3390/toxins11020110_

Round 1

Reviewer 1 Report

The paper reports an interesting longitudinal study on the presence of AFM1 in Pakistan commercial milk and milk-products. Laboratory and statistical analyses are well described and conducted. I appreciated this paper and I suggest the publication after minor revisions.

Lines 118-121 and lines 193-194. Adulterants/stabilizers analyses and results appear only partially justified and poorly described. Please define aims and procedures. 

Line 187. Please give evidence of this sentence; explain what adulteration stands for and why this analysis is useful for the aim of the present survey

Lines 151-157. Conclusion paragraph: Please add some more considerations about the significance of your study. Please define the innovation aspects of your results .

Ref n° 13. the paper is now published, please cite it correctely

In table 1, please  specify n of samples collected for each period for each type of matrix

In table 1 please verify the footnote n° 1: is not reported inside the table

Author Response

We highly appreciate the time taken by reviewers and editorial office in suggesting potential improvements in our manuscript. We have sincerely tried to revise the manuscript according to the suggestions, and a revised manuscript is being submitted accordingly.

Kindly consider the following reply to reviewer comments. We would like to inform that we will welcome any further suggestions in this regard.

Suggestion 1: Lines 118-121 and lines 193-194. Adulterants/stabilizers analyses and results appear only partially justified and poorly described. Please define aims and procedures.

Reply: Thank you for pointing out this problem. Aims of adulteration analysis are now explained in lines 187-189 which should address the suggestion. Some points have also been added from Lines 113 to 122.

Overall, the paper is aimed at AFM1 analysis and adulteration was only included to identify any notable case of milk dilution. Adulteration with non-milk constituents or water would dilute the milk and therefore comparison of adulterated milk with pure milk is not justified.

Regarding the procedures: we are of the view that the qualitative tests are not the main aim of this study and the given reference, freely available on internet, describes the methods in detail. Therefore further description will be a duplication of the already printed matter. Please do not hesitate to notify if we should further describe the methods.

Suggestion 2: Line 187. Please give evidence of this sentence; explain what adulteration stands for and why this analysis is useful for the aim of the present survey

Reply: Revised as suggested.

Suggestion 3: Lines 151-157. Conclusion paragraph: Please add some more considerations about the significance of your study. Please define the innovation aspects of your results

Reply: As mentioned in lines 49-51, present study is the first longitudinal study in Pakistan that includes all types of milk available in the market. We agree that conclusions could have included more points. However, we are very unfortunately restricted by the funding source and Federal Ministry to not go further in the conclusions.

Suggestion 4Ref n° 13. the paper is now published, please cite it correctly

Reply: Revised as suggested.

Suggestion 5: In table 1, please  specify n of samples collected for each period for each type of matrix

Reply: Total number of samples for each matrix are mentioned in the table. We are of the view that mentioning n values with each data point will make the table unfriendly for the reader. If it is important, then we humbly suggest that the table with n values with each data point may additionally be given as a supplement.

Suggestion 6: In table 1 please verify the footnote n° 1: is not reported inside the table

Reply: Revised as suggested.

Reviewer 2 Report

The paper gives information that can be useful for risk managers of the geographical area where the study was performed.

Only some little improvements are required, as suggested below

Introduction

Line 24: the importance of milk could be emphasized more than that of eggs, as at present the entity of theese two food matrices seems to be quantitatively different.

Line 31: a difference in mycotoxin threshold in EU and USA laws is cited; an example of this difference (e.g. half cncentration in UE legislation)

Results and discussion

Figure 1, line 64: "0.021" Is it "0.001"?

Line 76: "four"

Line 106: "two"

Line 111: "AFM1/L limit."

Materials and methods

Line 187: Are there data detecting a frequent adulteration? This sentence can be clearly justified?

Lines 200-201: the milk consumption by the 5 years old children was taken from the manufacturer's recommendation, as the value for one year old children?

References

Line 216: "Pakistan" not Italics

Line 228: "Food Control" (No official abbreviation)

Line 230 "Pakistan" not Italics

Author Response

We highly appreciate the time taken by reviewers and editorial office in suggesting potential improvements in our manuscript. We have sincerely tried to revise the manuscript according to the suggestions, and a revised manuscript is being submitted accordingly.

Kindly consider the following reply to reviewer comments. We would like to inform that we will welcome any further suggestions in this regard.

Suggestion 1: Introduction - Line 24: the importance of milk could be emphasized more than that of eggs, as at present the entity of these two food matrices seems to be quantitatively different.

Reply: Revised as suggested.

Suggestion 2: Line 31: a difference in mycotoxin threshold in EU and USA laws is cited; an example of this difference (e.g. half concentration in UE legislation).

Reply: Revised as suggested.

Suggestion 3: Figure 1, line 64: "0.021" Is it "0.001"?

Reply: Revised to be P < 0.05 both at suggested site and in abstract.

Suggestion 4: Line 76: "four"

Reply: Revised as suggested.

Suggestion 5: Line 106: "two"

Reply: Revised as suggested.

Suggestion 6: Line 111: "AFM1/L limit."

Reply: Revised as suggested.

Suggestion 7: Line 187: Are there data detecting a frequent adulteration? This sentence can be clearly justified?

Reply: Revised as suggested.

Suggestion 8: Lines 200-201: the milk consumption by the 5 years old children was taken from the manufacturer's recommendation, as the value for one year old children?

Reply: Each milk powder brand was for different age group of children as mentioned in Line 164. Recommended daily consumption by manufacturers is same probably due to the fact that 5 year old children derive nutrients from other stuffs.

Suggestion 9: Line 216: "Pakistan" not Italics.

Revised as suggested – for Line 217.

Suggestion 10: Line 228: "Food Control" (No official abbreviation)

Reply: Revised as suggested.

Suggestion 11: Line 230 "Pakistan" not Italics

Reply: Revised as suggested.